# Adenosine Improves Mitochondrial Function and Biogenesis in Friedreich’s Ataxia Fibroblasts Following L-Buthionine Sulfoximine-Induced Oxidative Stress

**DOI:** 10.3390/biology12040559

**Published:** 2023-04-06

**Authors:** Sze Yuen Lew, Nur Shahirah Mohd Hisam, Michael Weng Lok Phang, Syarifah Nur Syed Abdul Rahman, Rozaida Yuen Ying Poh, Siew Huah Lim, Mohd Amir Kamaruzzaman, Sze Chun Chau, Ka Chun Tsui, Lee Wei Lim, Kah Hui Wong

**Affiliations:** 1Department of Anatomy, Faculty of Medicine, Universiti Malaya, Kuala Lumpur 50603, Malaysia; 2Department of Oral and Craniofacial Sciences, Faculty of Dentistry, Universiti Malaya, Kuala Lumpur 50603, Malaysia; 3Department of Biomedical Science, Faculty of Medicine, Universiti Malaya, Kuala Lumpur 50603, Malaysia; 4Department of Chemistry, Faculty of Science, Universiti Malaya, Kuala Lumpur 50603, Malaysia; 5Department of Anatomy, Faculty of Medicine, Universiti Kebangsaan Malaysia, Jalan Yaacob Latiff, Cheras, Kuala Lumpur 56000, Malaysia; 6School of Biomedical Sciences, Li Ka Shing Faculty of Medicine, The University of Hong Kong, Hong Kong SAR, China

**Keywords:** adenosine, Friedreich’s ataxia, dermal fibroblasts, oxidative stress, mitochondrial function, mitochondrial biogenesis

## Abstract

**Simple Summary:**

Friedreich’s ataxia is the most common form of inherited ataxia, with an estimated prevalence of 1:50,000 in Caucasians. With no cure and a reduced lifespan, Friedreich’s ataxia is a devastating neurodegenerative disease. At the present time, the treatment strategies are aimed at specific symptoms, such as supportive treatment and physical therapy for motor dysfunction. Medication could improve the symptoms. However, side effects may include intolerable nausea, insomnia, and/or depression. The lack of effective therapeutic options remains a major gap in the field. Mitochondrial dysfunction and oxidative stress have been implicated in the pathogenesis of Friedreich’s ataxia. Here, we investigated the protective effects of adenosine against mitochondrial impairment in Friedreich’s ataxia. We showed that adenosine attenuated the deleterious effects of oxidative stress and mitochondrial dysfunction by regulating mitochondrial function and biogenesis in fibroblasts derived from a Friedreich’s ataxia patient. It serves as a promising therapeutic associated with mitochondrial dynamics that could eventually be a major breakthrough in the treatment of Friedreich’s ataxia, ultimately improving the quality of life of Friedreich’s ataxia patients and their caregivers and reducing its associated healthcare burden.

**Abstract:**

Adenosine is a nucleoside that is widely distributed in the central nervous system and acts as a central excitatory and inhibitory neurotransmitter in the brain. The protective role of adenosine in different pathological conditions and neurodegenerative diseases is mainly mediated by adenosine receptors. However, its potential role in mitigating the deleterious effects of oxidative stress in Friedreich’s ataxia (FRDA) remains poorly understood. We aimed to investigate the protective effects of adenosine against mitochondrial dysfunction and impaired mitochondrial biogenesis in L-buthionine sulfoximine (BSO)-induced oxidative stress in dermal fibroblasts derived from an FRDA patient. The FRDA fibroblasts were pre-treated with adenosine for 2 h, followed by 12.50 mM BSO to induce oxidative stress. Cells in medium without any treatments or pre-treated with 5 µM idebenone served as the negative and positive controls, respectively. Cell viability, mitochondrial membrane potential (MMP), aconitase activity, adenosine triphosphate (ATP) level, mitochondrial biogenesis, and associated gene expressions were assessed. We observed disruption of mitochondrial function and biogenesis and alteration in gene expression patterns in BSO-treated FRDA fibroblasts. Pre-treatment with adenosine ranging from 0–600 µM restored MMP, promoted ATP production and mitochondrial biogenesis, and modulated the expression of key metabolic genes, namely nuclear respiratory factor 1 (*NRF1*), transcription factor A, mitochondrial (*TFAM*), and NFE2-like bZIP transcription factor 2 (*NFE2L2*). Our study demonstrated that adenosine targeted mitochondrial defects in FRDA, contributing to improved mitochondrial function and biogenesis, leading to cellular iron homeostasis. Therefore, we suggest a possible therapeutic role for adenosine in FRDA.

## 1. Introduction

Friedreich’s ataxia (FRDA) is a degenerative autosomal recessive cerebellar ataxia, causing movement disorder. Patients with FRDA suffer from progressive gait and limb ataxia, weakness of the lower limbs, lack of tendon reflexes in the legs, dysarthria, hypertrophic cardiomyopathy, scoliosis, diabetes mellitus, and skeletal deformities [1]. Early-onset patients often become wheelchair-bound at a median of 11.5 years after onset [2]. These patients have a reduced life expectancy. Cardiac dysfunctions, including dilated cardiomyopathy and arrhythmia, are the most common cause of death in FRDA [3]. FRDA is the most common inherited cerebellar ataxia in individuals of Western European origin, with a prevalence between 1:20,000 and 1:725,000. It is also found in those of North African and Middle Eastern origin, but it has not been reported in other ethnic groups [4].

FRDA is caused by the expansion of guanine–adenine–adenine (GAA) trinucleotide repeats in the first intron of the frataxin (*FXN*) gene on chromosome 9q21.11, leading to substantially decreased levels of mitochondrial protein *FXN* [5]. Reduced *FXN* level causes increased cellular oxidative stress and impairs the formation of iron–sulfur (Fe-S) clusters such as heme, electron transport chain complexes I-III, and aconitase of the Krebs cycle [6]. Frataxin deficiency has been reported to impair the regulation of iron in the formation of Fe-S clusters, resulting in iron accumulation in mitochondria [7,8,9]. Excessive iron accumulation in the mitochondrial matrix causes the generation of reactive oxygen species (ROS) via the Fenton reaction and, therefore, increases oxidative stress and inactivates mitochondrial enzymes [10,11]. Disrupted mitochondrial enzyme activity leads to decreased mitochondrial respiration and production of adenosine triphosphate (ATP), resulting in mitochondrial energy imbalance and mitochondrial dysfunction [12,13].

Adenosine is a nucleoside that is widely distributed in the central nervous system (CNS) and acts as a central excitatory and inhibitory neurotransmitter in the brain [14]. It has multiple roles, including regulation of neurotransmitter release from synaptic vesicles [15,16], neuronal hyperpolarization or depolarization [17,18], and glial cell activity [19]. Increased production of extracellular adenosine has been found to play a distinct role in intercellular signaling by engaging cell surface adenosine receptors during myocardial ischemia–reperfusion injury and hypoxia and inflammation in acute respiratory distress syndrome and chronic lung diseases [14]. Although short-lived in circulation, adenosine exerts its action through activation of specific G-protein-coupled receptors, for which four subtypes (adenosine A_1_ receptor_,_ A_1_R; adenosine A_2A_ receptor, A_2A_R; adenosine A_2B_ receptor, A_2B_R; and adenosine A_3_ receptor, A_3_R) have been identified so far [20,21]. However, interaction with adenosine receptors depends on disease type, location of lesion, and distribution of receptors [14]. Several CNS diseases, including cerebral ischemia [22], Alzheimer’s disease, depression [23], and epilepsy, are associated with decreased expression of A_1_R and increased expression of A_2A_R [24]. Kao et al. [25] and Lee et al. [26] also postulated that reduced levels of adenosine may cause neurological impairments in Huntington’s and Alzheimer’s disease, respectively.

To date, only one A_2A_R antagonist, namely istradefylline (KW6002), has been approved by the US Food Drug and Administration (FDA) as an adjuvant treatment with levodopa or carbidopa for patients with Parkinson’s disease [27,28]. Nevertheless, the therapeutic efficacy of adenosine-related treatments for many neurological and neurodegenerative diseases remains inconclusive [29].

In the past two decades, there has been scant evidence elucidating the protective effects of adenosine against cerebellar ataxia. Heffer et al. [30] showed that behavioral profiling of adenosine agonists is related to their affinity for A_1_R and A_2_R, similar to that of dopamine antagonists in the attenuation of spontaneous locomotor activity in mice. On the other hand, the cerebellar adenosinergic system involving A_1_R had been shown to promote ethanol- [31] and cannabinoid-induced [32,33] motor incoordination, whereas A_1_R antagonist attenuated the effect, suggesting modulation by an endogenous A_1_R. However, the mechanism of accentuation of motor incoordination by *N^6^*-cyclohexyladenosine, an A_1_R-selective agonist, was not investigated in the study. Moreover, Dar and Mustafa [34] found that antisense oligodeoxynucleotide targeting A_1_R significantly decreased the protein level of A_1_R.

The current study aims to bridge the research gap between the reported therapeutic effects of adenosine and the involvement of mitochondrial function and biogenesis in FRDA. Here, we investigated the protective effects of adenosine against mitochondrial dysfunction and impaired mitochondrial biogenesis in L-buthionine sulfoximine (BSO)-induced oxidative stress in dermal fibroblasts derived from an FRDA patient.

## 2. Materials and Methods

### 2.1. Chemicals and Reagents

Reagents used in this study include Dulbecco’s Modified Eagle Medium (DMEM) (Gibco, Thermo Fisher Scientific, Waltham, MA, USA); fetal bovine serum (FBS) (Biosera, Nuaillé, France); idebenone, L-buthionine sulfoximine (BSO), and aconitase assay kit (Cayman Chemical, Ann Arbor, MI, USA); adenosine, penicillin–streptomycin, and mitochondrial membrane potential (MMP) kit (Sigma-Aldrich, St. Louis, MO, USA); 3-(4,5-dimethylthiazol-2-yl)-2,5-diphenyltetrazolium bromide (MTT) (Alfa Aesar, Haverhill, MA, USA); Mitochondrial ToxGlo^TM^ assay kit (Promega Corporation, Madison, WI, USA); MitoBiogenesis™ In-Cell ELISA Kit (Abcam, Cambridge, UK); TRIzol^®^ reagent (Life Technologies, Thermo Fisher Scientific, Waltham, MA, USA); RevertAid First Strand cDNA Synthesis Kit (Thermo Fisher Scientific, Waltham, MA, USA); qPCRBIO SyGreen Mix Separate-ROX (PCR Biosystems, London, UK); and OmniPur^®^ water (Merck Millipore, Darmstadt, Germany).

### 2.2. Dermal Fibroblasts Culture

Dermal fibroblasts from a healthy 33-year-old male (GM02673) and dermal fibroblasts from a 30-year-old FRDA male patient (GM04078) were purchased from Coriell Institute for Medical Research (Camden, NJ, USA) under the terms of the Material Transfer Agreement (Assurance Form) with the Universiti Malaya, with Dr. Kah Hui Wong as the principal investigator. Both dermal fibroblasts from the National Institute of General Medical Sciences (NIGMS) Human Genetic Cell Repository have been collected under Institutional Review Board (IRB) approval, and patient informed consent. The species of origin and molecular characterization of the fibroblasts were confirmed by Coriell Institute for Medical Research via microsatellite analysis and nucleoside phosphorylase isoenzyme electrophoresis. The FRDA fibroblasts are homozygous for GAA expansion in the *FXN* gene with alleles of approximately 541 and 420 repeats. Dermal fibroblasts were maintained in DMEM supplemented with 15% (*v*/*v*) FBS and 1% (*v*/*v*) penicillin–streptomycin at 37 ± 2 °C in a 5% CO_2_-humidified incubator according to a previous protocol [35]. Fibroblasts in DMEM without any treatment or pre-treated with 5 μM idebenone [36] served as the negative and positive controls, respectively. Idebenone, an analog of coenzyme Q_10_, has been used as an antioxidant therapy for the management of hypertrophic cardiomyopathy in patients with FRDA [37]. Prior to the assays, the medium was changed to phenol red-free and sodium pyruvate-free DMEM.

### 2.3. 3-(4,5-dimethylthiazol-2-yl)-2,5-diphenyltetrazolium Bromide (MTT) Viability Assay

The normal and FRDA fibroblasts were plated at a density of 1 × 10^4^ cells per well in a 96-well plate and incubated for 24 h at 37 ± 2 °C in a 5% CO_2_-humidified incubator. The medium was replaced with a fresh medium containing adenosine ranging from 0 to 2000 µM for 24 h. Fibroblasts in DMEM without any treatment served as the negative control. After incubation, 0.5 mg/mL MTT solution was added to each well and incubated for 4 h at 37 ± 2 °C in a 5% CO_2_-humidified incubator. The reduction of MTT from a yellow tetrazolium dye to purple formazan crystals can be used to measure cellular enzymes in viable cells. The medium was then removed, and dimethyl sulfoxide (DMSO) was added to each well. Absorbance was measured at 570 nm with 630 nm as background absorbance using a multimode microplate reader (SpectraMax^®^ M3, Molecular Devices, Union City, CA, USA). The cell viability was expressed as a percentage of the negative control level.

As FRDA fibroblasts are lacking in frataxin, the cells are extremely sensitive to BSO-induced oxidative stress compared to normal fibroblasts. Frataxin has been shown to be influential in the production of Fe-S cluster-containing proteins [35]. We observed a dose-response curve of viability of normal and FRDA fibroblasts following exposure to varying concentrations of BSO for 24 h. The viability of FRDA fibroblasts decreased gradually as the concentration of BSO increased from 1.56 to 6.25 mM, with a sharp decrease in viability at 12.50 mM BSO. Large-scale death of FRDA fibroblasts was markedly pronounced by challenging the cells with 12.50 mM BSO, by which the viability was reduced to 37.47 ± 0.23%. On the other hand, 12.50 mM BSO produced mild cytotoxicity effects in normal fibroblasts with a viability of 84.81 ± 5.62% [38]. As 12.50 mM BSO produced more than 50% reduction (*p * < 0.05) in the viability of FRDA fibroblasts, the concentration was selected for the subsequent assays of mitochondrial function and mitochondrial biogenesis to induce oxidative stress in FRDA fibroblasts. Normal fibroblasts were not tested for these assays.

### 2.4. Mitochondrial Membrane Potential (MMP) Assay

The FRDA fibroblasts were plated at a density of 1 × 10^4^ cells per well in a 96-well black plate and incubated for 24 h at 37 ± 2 °C in a 5% CO_2_-humidified incubator. The medium was replaced with a fresh medium containing adenosine ranging from 0 to 800 µM or 5 µM idebenone and incubated for 2 h, followed by 12.50 mM BSO [38] for 24 h. Fibroblasts in DMEM without any treatment or pre-treated with 5 μM idebenone [36] served as the negative and positive controls, respectively. The MMP was quantified in terms of the permeability of JC-10 dye using the MMP kit according to the manufacturer’s protocol. Fluorescence intensities were measured using a multimode microplate reader (SpectraMax^®^ M3, Molecular Devices, Union City, CA, USA) at 540 and 590 nm (red fluorescence) and 490 and 525 nm (green fluorescence) as the excitation and emission wavelengths, respectively. The ratio of red to green fluorescence intensity was expressed as a percentage of the negative control level.

### 2.5. Aconitase Assay

The FRDA fibroblasts were plated at a density of 1 × 10^5^ cells per well in a 6-well plate and incubated for 24 h at 37 ± 2 °C in a 5% CO_2_-humidified incubator. The medium was replaced with a fresh medium containing adenosine ranging from 0 to 600 µM or 5 µM idebenone and incubated for 2 h, followed by 12.50 mM BSO [38] for 24 h. Fibroblasts in DMEM without any treatment or pre-treated with 5 μM idebenone [36] served as the negative and positive controls, respectively. Fibroblasts were homogenized in an assay buffer and centrifuged at 800× *g* for 10 min at 4 °C to obtain the supernatant. Aconitase activity was determined using the aconitase assay kit according to the manufacturer’s protocol. Absorbance was measured at 450 nm using a multimode microplate reader (SpectraMax^®^ M3, Molecular Devices, Union City, CA, USA). The activity was expressed as a percentage of the negative control level.

### 2.6. Adenosine Triphosphate (ATP) Assay

The FRDA fibroblasts were plated at a density of 1 × 10^4^ cells per well in a 96-well white plate and incubated for 24 h at 37 ± 2 °C in a 5% CO_2_-humidified incubator. The medium was replaced with a fresh medium containing adenosine ranging from 0 to 600 µM or 5 µM idebenone and incubated for 2 h, followed by 12.50 mM BSO [38] for 24 h. Fibroblasts in DMEM without any treatment or pre-treated with 5 μM idebenone [36] served as the negative and positive controls, respectively. The ATP content was determined using the Mitochondrial ToxGlo^TM^ assay kit according to the manufacturer’s protocol. Luminescence signal was measured using a multimode microplate reader (SpectraMax^®^ M3; Molecular Devices, Union City, CA, USA). The activity was expressed as a percentage of the negative control level.

### 2.7. Mitochondrial Biogenesis Assay

The FRDA fibroblasts were plated at a density of 1 × 10^4^ cells per well in a 96-well plate and incubated for 24 h at 37 ± 2 °C in a 5% CO_2_-humidified incubator. The medium was replaced with a fresh medium containing adenosine ranging from 0 to 600 µM or 5 µM idebenone and incubated for 2 h, followed by 12.50 mM BSO [38] for 24 h. Fibroblasts in DMEM without any treatment or pre-treated with 5 μM idebenone [36] served as the negative and positive controls, respectively. Mitochondrial biogenesis was determined using the MitoBiogenesis™ In-Cell ELISA Kit according to the manufacturer’s protocol. Absorbance was measured at 405 and 650 nm for COX1 and SDH-A, respectively, using a multimode microplate reader (SpectraMax^®^ M3; Molecular Devices, Union City, CA, USA). The activity was expressed as a ratio of SDH-A to COX1.

### 2.8. Reverse Transcription Quantitative Real-Time Polymerase Chain Reaction (RT-qPCR)

The FRDA fibroblasts were plated at a density of 1 × 10^5^ cells per well in a 6-well plate and incubated for 24 h at 37 ± 2 °C in a 5% CO_2_-humidified incubator. The medium was replaced with a fresh medium containing adenosine ranging from 0 to 600 µM or 5 µM idebenone and incubated for 2 h, followed by 12.50 mM BSO [38] for 24 h. Fibroblasts in DMEM without any treatment or pre-treated with 5 μM idebenone [36] served as the negative and positive controls, respectively. Fibroblasts were subjected to total RNA extraction using TRIzol^®^ reagent according to the manufacturer’s protocol. The concentration of total RNA was determined using the NanoDrop™ 2000/2000c Spectrophotometers (Thermo Fisher Scientific, Waltham, MA, USA). Approximately 300 ng of total RNA was converted to cDNA using RevertAid First Strand cDNA Synthesis Kit according to the manufacturer’s protocol and Veriti^®^ 96-Well Thermal Cycler (Applied Biosystems Inc., Foster City, CA, USA). The RT-qPCR was performed using 15 ng/µL cDNA template in a pooled solution containing qPCRBIO SyGreen Mix Separate-ROX, 10 pmol/µL forward and reverse oligonucleotide primers, and OmniPur^®^ water. The primer sequences used in this study were synthesized by Genewiz (South Plainfield, NJ, USA), as shown in Table 1. Amplification was performed for 40 cycles in the StepOne Plus™ Real-Time PCR System (Applied Biosystems Inc., Foster City, CA, USA) [39,40,41]. The expression of the gene of interest was normalized to the reference gene actin beta (*ACTB*). Fold change of gene expression was determined using the comparative threshold cycle method (ΔΔCt) [42,43].

### 2.9. Statistical Analysis 

All statistical analyses were performed in Statistical Package for the Social Science (SPSS) 22.0, and data were presented as mean ± standard deviation (SD) from three independent biological replicates (*n* = 3). The Shapiro–Wilk test was employed to evaluate the normality of the data. Normally distributed data from more than two experimental groups were examined by Levene’s test to evaluate the homogeneity of variances between groups. All groups with equal variances assumed were evaluated by one-way analysis of variance (ANOVA) to examine differences between groups, followed by Tukey’s honestly significant difference (HSD) post-hoc test. All groups with equal variances not assumed were evaluated by ANOVA, followed by Games–Howell multiple comparison post-hoc test. A statistical difference of *p* < 0.05 was considered significant.

## 3. Results

### 3.1. Effects of Adenosine on the Viability of Normal and FRDA Fibroblasts

Prior to the investigation of the protective effects of adenosine, the viability of adenosine-treated normal and FRDA fibroblasts was determined to exclude possible cytotoxic and proliferative effects. Fibroblasts were exposed to various concentrations of adenosine. As shown in Figure 1, the viability of normal and FRDA fibroblasts gradually decreased with increasing concentration of adenosine from 0 to 2000 µM. As the relatively lower concentrations of 0 to 800 µM adenosine showed no significant difference in the viability compared to the negative control (*p* > 0.05), these concentrations were selected to investigate its protective effects against mitochondrial dysfunction and impaired mitochondrial biogenesis in FRDA fibroblasts challenged with 12.50 mM BSO.

### 3.2. Effects of Adenosine on the Mitochondrial Membrane Potential (MMP) in FRDA Fibroblasts Treated with BSO

The MMP plays an important part in the energy storage process during oxidative phosphorylation. We used JC-10 dye to discriminate energized and de-energized mitochondria, indicated by red fluorescent aggregates in energized mitochondria with increasing membrane potential and by green fluorescence monomers in cells with mitochondria with collapsed MMP that fail to retain the dye. As shown in Figure 2, FRDA fibroblasts treated with 12.50 mM BSO showed significantly decreased MMP of 87.86 ± 2.8% or 1.1-fold lower compared to the negative control (*p* < 0.05). However, pre-treatment with 0 to 800 µM adenosine significantly increased MMP (104.77 ± 3.0%, 101.94 ± 4.4%, 100.66 ± 0.7%, and 101.77 ± 2.9%, respectively) to 1.1- to 1.2-fold higher compared to BSO (*p* < 0.05). All tested concentrations of adenosine exhibited 1.2- to 1.3-fold higher MMP compared to idebenone (*p* < 0.05). As the relatively lower concentrations of 0 to 600 µM adenosine showed higher MMP compared to 800 µM adenosine, these concentrations were selected in the subsequent assays of mitochondrial function and biogenesis.

### 3.3. Effects of Adenosine on the Aconitase Activity in FRDA Fibroblasts Treated with BSO

Aconitase is an Fe-S protein in the Krebs cycle that catalyzes the isomerization of citrate to isocitrate. As shown in Figure 3, FRDA fibroblasts treated with 12.50 mM BSO showed significantly decreased aconitase activity of 59.33 ± 11.2% or 1.7-fold lower compared to the negative control (*p* < 0.05). Pre-treatment with adenosine ranging from 0 to 600 µM failed to restore the depleted aconitase activity (*p* > 0.05). However, pre-treatment with 400 and 600 µM adenosine resulted in 1.4- to 1.5-fold higher activity compared to idebenone (*p* > 0.05).

### 3.4. Effects of Adenosine on the Adenosine Triphosphate (ATP) Level in FRDA Fibroblasts Treated with BSO

The ATP is the source of energy for use and storage at the cellular level. As shown in Figure 4, FRDA fibroblasts treated with 12.50 mM BSO showed a significantly decreased ATP level of 74.83 ± 3.2% or 1.3-fold lower compared to the negative control (*p* < 0.05). However, pre-treatment with 200 µM adenosine increased the ATP level to 85.45 ± 4.7% or 1.1-fold higher compared to BSO (*p* < 0.05). All tested concentrations of adenosine exhibited 179.1- to 203.5-fold higher ATP levels compared to idebenone (*p* < 0.05).

### 3.5. Effects of Adenosine on the Mitochondrial Biogenesis in FRDA Fibroblasts Treated with BSO

Mitochondrial biogenesis is the growth and division of pre-existing mitochondria. In this study, mitochondrial biogenesis was evaluated by the ratio of two mitochondrial proteins, namely succinate dehydrogenase subunit A (SDH-A), a subunit of complex II (nuclear DNA, nDNA-encoded) and cytochrome c oxidase subunit 1 (COX1), a subunit of complex IV (mitochondrial DNA, mtDNA-encoded). As shown in Figure 5, there was no significant difference in the SDH-A/COX1 ratio between negative control and FRDA fibroblasts treated with 12.50 mM BSO (*p* > 0.05). However, pre-treatment with 400 and 600 µM adenosine significantly increased the ratio to 109.77 ± 3.5% and 108.32 ± 4.6% or 1.2-fold higher compared to BSO, respectively (*p* < 0.05). All tested concentrations of adenosine exhibited a 2.1- to 2.2-fold higher SDH-A/COX1 ratio compared to idebenone (*p* < 0.05).

### 3.6. Effects of Adenosine on the Gene Expression Associated with Mitochondrial Biogenesis in FRDA Fibroblasts Treated with BSO

The *NRF1* modulates mitochondrial biogenesis by binding to specific promoter sites and regulating the expression of *TFAM.* As shown in Figure 6A, FRDA fibroblasts treated with 12.50 mM BSO showed significantly increased expression of *NRF1* or 2.6-fold higher compared to the negative control (*p* < 0.05). However, pre-treatment with 400 and 600 µM adenosine significantly decreased the expression of *NRF1* to 1.3- and 1.5-fold lower compared to BSO, respectively (*p* < 0.05). All tested concentrations of adenosine exhibited 8.8- to 11.6-fold lower expression of *NRF1* compared to that of idebenone (*p* < 0.05).

The *TFAM*, an nDNA-encoded mtDNA-binding protein, is required for the regulation of mtDNA transcription, replication, and maintenance. As shown in Figure 6B, FRDA fibroblasts treated with 12.50 mM BSO significantly increased the expression of *TFAM* or 3.1-fold higher compared to the negative control (*p* < 0.05). However, pre-treatment with 400 and 600 µM adenosine significantly decreased the expression of *TFAM* to 1.3- and 1.5-fold lower compared to BSO, respectively (*p* < 0.05). All tested concentrations of adenosine exhibited 4.0- to 4.6-fold lower expression of *TFAM* compared to that of idebenone (*p* < 0.05).

The *PPARGC1A* is the master regulator of mitochondrial biogenesis and energy expenditure. It modulates the activity of transcription factors, including *NRF1*. As shown in Figure 6C, there was no significant difference in the expression of *PPARGC1A* between the negative control and FRDA fibroblasts treated with 12.50 mM BSO (*p* > 0.05). Pre-treatment with adenosine ranging from 0 to 600 µM did not decrease the expression of *PPARGC1A* compared to BSO (*p* > 0.05). All tested concentrations of adenosine exhibited 20.6- to 37.7-fold lower expression of *PPARGC1A* compared to that of idebenone (*p* < 0.05).

The *NFE2L2* is a major transcription factor that regulates the expression of various antioxidant genes, redox homeostasis, and anti-inflammatory and metabolic enzymes. Additionally, *NFE2L2*/antioxidant response element (ARE) signaling cascade has been observed to activate mitochondrial biogenesis. As shown in Figure 6D, FRDA fibroblasts treated with 12.50 mM BSO significantly increased the expression of *NFE2L2* or 13.5-fold higher compared to the negative control (*p* < 0.05). However, pre-treatment with 400 and 600 µM adenosine significantly decreased the expression of *NFE2L2* to 1.5- and 79.6-fold lower compared to BSO, respectively (*p* < 0.05). Adenosine (600 µM) exhibited a 21.2-fold lower expression of *NFE2L2* compared to that of idebenone (*p* < 0.05).

## 4. Discussion

In this study, we investigated the protective effects of adenosine against BSO-induced oxidative stress in an in vitro model of FRDA using fibroblasts derived from an FRDA patient. Our aim was to elucidate the mechanism of action in which adenosine can exert its antioxidant potential through the involvement of mitochondrial function and biogenesis and associated gene expression patterns following oxidative stress. 

The key pathophysiological features of FRDA include excessive iron accumulation in the mitochondrial matrix, mitochondrial dysfunction, mitochondrial energy imbalance due to decreased ATP production, and increased sensitivity to oxidative stress [12]. Currently, there is no FDA-approved treatment to halt the progression of FRDA [44]. Various pharmacological agents and natural products targeting oxidative stress [45,46,47,48,49,50,51] have been considered in the management of neuromuscular disorders, including FRDA [38,52,53]. Adenosine and its derivatives have been reported to exhibit free radical scavenging activities [54] and protect bone marrow-derived neural stem cells against hydrogen peroxide (H_2_O_2_)-induced oxidative stress and apoptosis [55].

Based on in vitro and in vivo studies, idebenone has been widely portrayed as a potent antioxidant [56,57], facilitating the redox flux of the mitochondrial electron transport chain to generate cellular ATP through oxidative phosphorylation. Idebenone was assigned Orphan Drug Status (EU/3/04/189) by the European Commission on 8 March 2004 for the treatment of cognitive and behavioral deficits, FRDA, and Leber’s hereditary optic neuropathy (LHON). Further, idebenone reduced impaired respiratory function in patients with Duchenne muscular dystrophy [58]. The QS10, a metabolite of idebenone, has been found to act as a replacement for endogenous CoQ10 in CoQ10-deficient cells and as a nutraceutical support to bypass complex I deficiency in LHON and autosomal dominant optic atrophy [59,60]. Idebenone has been observed to act as an antioxidant in preventing lipid peroxidation and rupturing of mitochondrial membrane function and as an electron carrier in supporting mitochondrial function and promoting ATP production in pre-clinical models [37,61,62,63,64,65] and clinical trials [66,67,68,69,70] of FRDA.

We observed that adenosine ranging from 0 to 800 µM did not cause cytotoxicity in normal and FRDA fibroblasts. Such a conclusion was also reached by Barth et al. [71], Jennings et al. [72], and Qi et al. [15]. Adenosine up to 500 µM did not induce cytotoxicity in cultured organotypic hippocampal slices [71] and normal human dermal fibroblasts [72]. Qi et al. [15] found that the effect of adenosine was concentration-dependent and saturated at 500 µM in suppressing excitatory transmission in layer 4 of the rat barrel cortex. Interestingly, 100 μM adenosine has been shown to maintain inflammasome activity and increase the duration of the inflammatory response through cAMP/PKA/CREB/HIF-1α pathway and A_2A_R, respectively, in mice deficient in Il1r, Nlrp3, Asc, P2X7, and caspase-1 primed with lipopolysaccharide [73]. 

The relative MMP in living cells can be a direct measurement of mitochondrial function. Excessive ROS production induces rapid depolarization of MMP and damages mitochondrial proteins and enzymes, leading to impaired oxidative phosphorylation and disrupted ATP generation [74]. Adenosine is a purine metabolite essential for the synthesis of ATP, the main energy source for cellular metabolism [75]. We observed that FRDA fibroblasts treated with BSO demonstrated reduced MMP, aconitase activity, and ATP levels, which is in line with previous studies [35,76]. Interestingly, pre-treatment with adenosine increased the MMP and ATP levels in BSO-treated FRDA fibroblasts. Xu et al. [77] observed that 100 µM adenosine restored the MMP in Wistar rat cardiomyocytes upon exposure to H_2_O_2_. Olatunji et al. [78] also demonstrated that adenosine isolated from *C. cicadae* attenuated the dissipation of MMP in glutamate-induced oxidative stress in PC-12 cells. Furthermore, Kalogeris et al. [79] showed that co-incubation of 10 µM adenosine with 1 ng/mL tumor necrosis factor alpha (TNF-α) attenuated apoptosis and increased MMP and ATP levels in human microvascular endothelial cells (HMEC-1). Janier et al. [80] observed that adenosine increased ATP levels in the heart tissue of New Zealand White rabbits upon exposure to ischemia and reperfusion injuries. In contrast, adenosine did not increase the MMP in C57BL/6J mice oocytes during meiotic maturation [81]. 

Aconitase, an enzyme of the Krebs cycle localized in mitochondria, is a prototypical example of a multifunctional protein. It is involved in the metabolic regulation of iron for maintaining ROS homeostasis. Aconitase has been reported to be severely affected in FRDA [82]. Indeed, while the activity of the mitochondrial enzyme is determinant for the metabolic flux through the tricarboxylic acid in the mitochondria, the cytosolic counterpart of the aconitase is known to regulate the overall iron metabolism of mammalian cells tightly. Our results indicate that pre-treatment with adenosine failed to increase the aconitase activity following exposure to BSO. Likewise, Lian and Stringer [83] also observed that pre-treatment with glutamine, a precursor of glutathione, did not rescue impaired mitochondrial aconitase activity in the rat cortical astrocytes upon exposure to cortical spreading depression and fluorocitrate, an inhibitor of the Krebs cycle. Not surprisingly in view of the critical role of aconitase in cell metabolism, inactivation of mitochondrial aconitase through an oxidative post-translational mechanism may lead to severe loss of aconitase activity.

Mitochondrial proteins are encoded by either nDNA or mtDNA, with approximately 99% of proteins encoded solely by nDNA. Therefore, nDNA-encoded mitochondrial proteins are the major determinant of mitochondrial abundance and mtDNA copy number following oxidative stress [84]. In this study, we assessed mitochondrial biogenesis of the respiratory chain, which is dependent on subunits encoded by both nuclear and mtDNA genes [85,86]. There was no significant difference in the SDH-A/COX1 ratio between the negative control and FRDA fibroblasts treated with BSO, indicating BSO did not alter mitochondrial biogenesis. Similarly, Aquilano et al. [87] found that increased mitochondrial biogenesis was not affected by the depletion of glutathione (GSH) in SH-SY5Y cells exposed to BSO, as the level of mitochondrial proteins such as heat shock protein 60 (Hsp60), COX, and cytochrome c remained unchanged. Interestingly, adenosine significantly increased the SDH-A/COX1 ratio in FRDA fibroblasts treated with BSO, indicating mitochondrial biogenesis may involve nDNA-encoded mitochondrial proteins. Moreover, SDH-A acts as a mitochondrial mass marker, as it is correlated with mitochondrial porin or voltage-dependent anion channel, the most abundant mitochondrial protein found in the outer membrane [88,89]. Vincent et al. [90] observed that low expression of SDH-A resulted in low mitochondrial mass in myofibers derived from patients with myofibrillar myopathy. Kalogeris et al. [79] demonstrated that HMEC-1 cells incubated with adenosine for 1–8 days increased the levels of mitochondrial biogenesis markers and mediators, including *PPARGC1A*, *NFE2L2*, porin, and cytochrome c oxidase subunit 4 (COX4, nDNA-encoded) in a time-dependent manner. 

Several key factors, such as *NRF1*, *TFAM*, *PPARGC1A*, and *NFE2L2*, have been found to modulate mitochondrial biogenesis. The *PPARGC1A* and *NFE2L2* regulate mitochondrial biogenesis by activating *NRF1*, leading to enhanced expression of *TFAM*, which is necessary for the replication and transcription of mtDNA [91,92,93]. In this study, BSO-treated FRDA fibroblasts showed increased expression of *NRF1, TFAM*, and *NFE2L2* but not *PPARGC1A*. Therefore, we postulated that BSO may induce low-level oxidative stress in FRDA fibroblasts and, thus, activates mitochondrial biogenesis through an *NFE2L2* signaling cascade. Marmolino et al. [94] reported that the expression of *PPARGC1A* remained unchanged in FRDA primary fibroblasts following exposure to H_2_O_2_, possibly due to low-level and chronic oxidative stress induced by H_2_O_2_. Additionally, BSO depleted the level of GSH in murine embryonic fibroblasts but upregulated the level of *NFE2L2*, suggesting its crucial role in cellular adaptive mechanisms following oxidative stress [95]. Furthermore, Aquilano et al. [96] demonstrated that BSO-treated SH-SY5Y cells increased the gene expression and protein levels of *NFE2L2*, superoxide dismutase 2 (SOD2), and γ-glutamylcysteine ligase (γ-GCL). 

In the present study, adenosine modulated the expression of *NRF1*, *TFAM*, and *NFE2L2* and increased nDNA-encoded mitochondrial proteins involved in mitochondrial biogenesis. This is in line with the recent findings of Walsh et al. [97], demonstrating that *NFE2L2* activators reduced the protein levels of *NFE2L2* and glutathione reductase, whereas the combination of *NRF1* and *NFE2L2* activators reduced the protein levels of complex II, III, IV, and V in C2C12 myoblasts upon exposure to H_2_O_2_. Walsh et al. [97] postulated that both *NRF1* and *NFE2L2* work synergistically to regulate a transcriptional process to promote adaptive homeostasis in response to stressful stimuli. Accordingly, Min et al. [98] also found that adenosine-induced nuclear translocation of *NFE2L2* and increased the protein levels of ARE in primary rat microglial and BV2 cells. In contrast to our observation, Kalogeris et al. [79] demonstrated an elevation of protein levels of *PPARGC1A* and *NFE2L2* up to 1.5- to 4-fold higher compared to negative control in HMEC-1 cells following incubation with 1 µM adenosine for 2-5 days. In addition, they also revealed that adenosine could mediate the preservation of mitochondrial mass in HMEC-1 cells following TNF-α-induced apoptosis by increasing the expression of *PPARGC1A*. 

Adenosine is a metabolite generated in response to hypoxia, injury, or inflammation, eliciting its protective or damaging responses through adenylyl cyclase inhibitory A_1_R and A_3_R and the adenylyl cyclase stimulatory A_2A_R and A_2B_R. Our observation is in line with previous findings [99,100,101]. Enhanced levels of extracellular adenosine have been reported to activate cytoprotective signaling through A_1_R in mediating the modulation of coronary flow, heart rate and contraction, cardioprotection, inflammatory response, cell growth, and tissue remodeling [99]. On the other hand, Castro et al. [100] demonstrated the protective effects of intraarticular injections of liposomal adenosine or A_2_ agonist against mitochondrial oxidized material and ROS burden in a mouse model of obesity-induced osteoarthritis. In addition, adenosine regulated metabolic functions through receptor-dependent mechanisms involving A_1_R and A_2A_R, and the formation of ROS and reactive nitrogen species in response to neuroinflammation in mixed glial cells and an animal model of neuroinflammation induced by intracerebroventricular administration of lipopolysaccharide [101].

Our findings show that adenosine possessed protective effects against BSO-induced oxidative stress and cell death in FRDA fibroblasts. Adenosine reconstructed mitochondrial function and regulated mitochondrial biogenesis, contributing to mitochondrial ROS and cellular iron homeostasis. The *NRF1*, *TFAM*, and *NFE2L2* are the key regulators of mitochondrial biogenesis (Figure 7).

## 5. Conclusions

Our study demonstrated that adenosine targeted mitochondrial defects in FRDA, contributing to improved mitochondrial function and biogenesis, leading to cellular iron homeostasis. Therefore, we suggest a possible therapeutic role for adenosine in FRDA.

## Figures and Tables

**Figure 1 biology-12-00559-f001:**
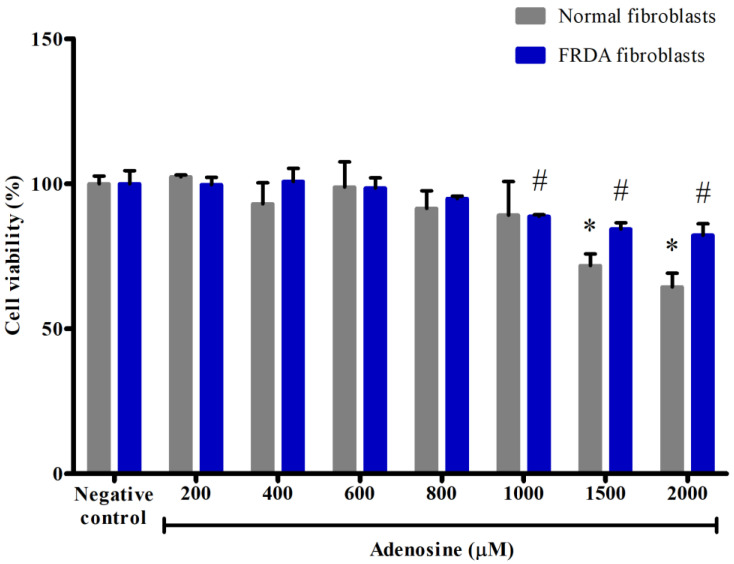
Effects of adenosine on the viability of normal and FRDA fibroblasts. Cell viability was evaluated by MTT assay following incubation with various concentrations of adenosine for 24 h. Data are expressed as mean ± SD and statistically analyzed by one-way ANOVA with Tukey’s post-hoc test. Asterisk (*) and hash sign (#) indicate significant differences (*p* < 0.05) in viability for different groups of normal and FRDA fibroblasts, respectively, compared to the negative control group.

**Figure 2 biology-12-00559-f002:**
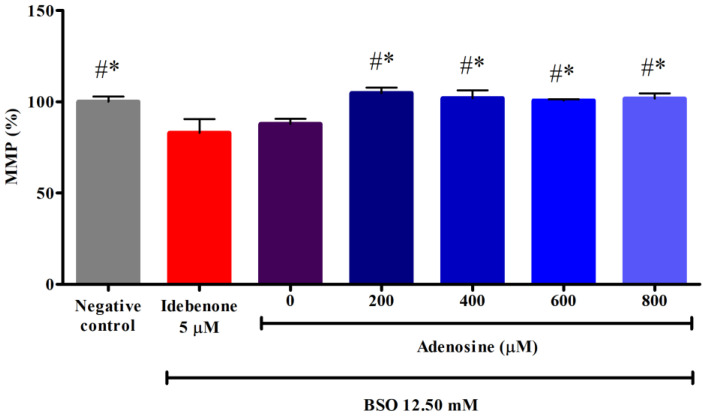
Effects of adenosine on the MMP in FRDA fibroblasts treated with BSO. The MMP was evaluated following pre-treatment with adenosine (0 to 800 µM) for 2 h and 12.50 mM BSO for 24 h. Data are expressed as mean ± SD and statistically analyzed by one-way ANOVA with Tukey’s post-hoc test. Hash (#) and asterisk (*) indicate significant differences (*p* < 0.05) in MMP compared to the idebenone group and BSO-treated (0 µM adenosine) group, respectively. BSO, L-buthionine sulfoximine; MMP, mitochondrial membrane potential.

**Figure 3 biology-12-00559-f003:**
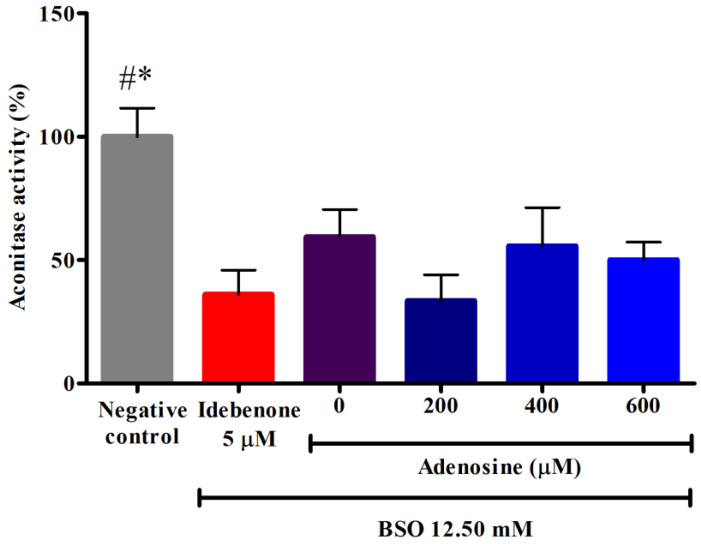
Effects of adenosine on the aconitase activity in FRDA fibroblasts treated with BSO. Aconitase activity was evaluated following pre-treatment with adenosine for 2 h and 12.50 mM BSO for 24 h. Data are expressed as mean ± SD and statistically analyzed by one-way ANOVA with Tukey’s post-hoc test. Hash (#) and asterisk (*) indicate significant differences (*p* < 0.05) in aconitase activity compared to the idebenone group and BSO-treated (0 µM adenosine) group, respectively. BSO, L-buthionine sulfoximine.

**Figure 4 biology-12-00559-f004:**
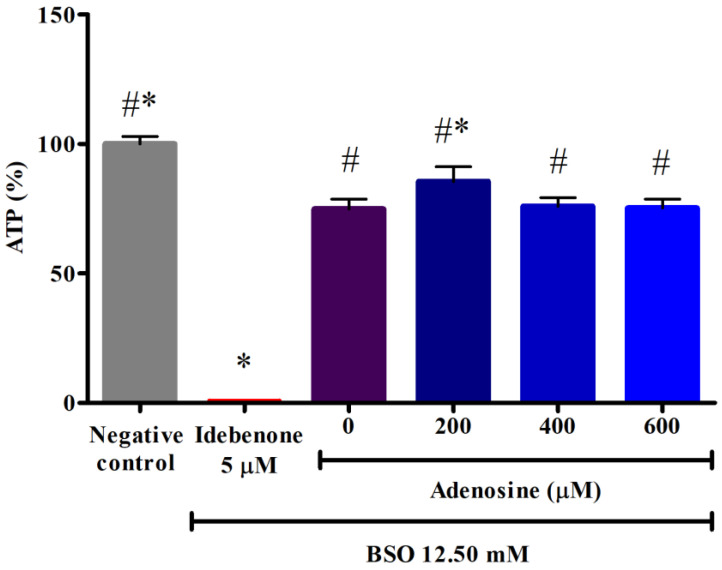
Effects of adenosine on the ATP level in FRDA fibroblasts treated with BSO. The ATP level was evaluated following pre-treatment with adenosine for 2 h and 12.50 mM BSO for 24 h. Data are expressed as mean ± SD and statistically analyzed by one-way ANOVA with Tukey’s post-hoc test. Hash (#) and asterisk (*) indicate significant differences (*p* < 0.05) in ATP level compared to the idebenone group and BSO-treated (0 µM adenosine) group, respectively. ATP, adenosine triphosphate; BSO, L-buthionine sulfoximine.

**Figure 5 biology-12-00559-f005:**
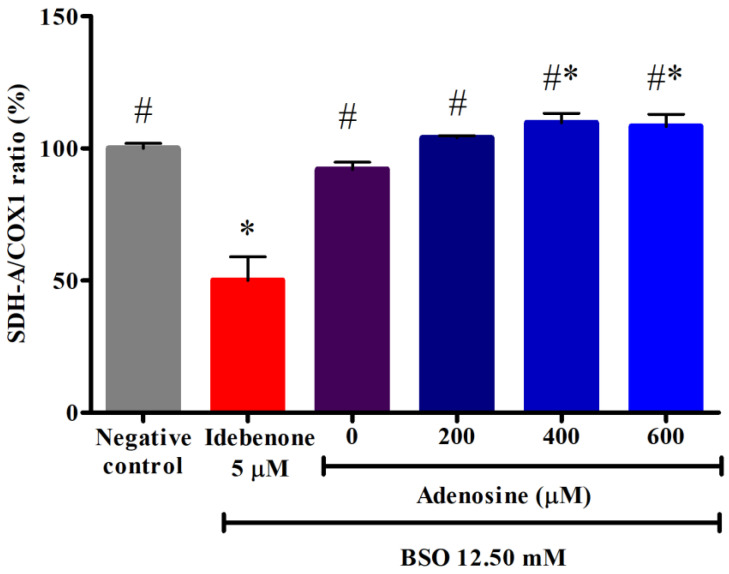
Effects of adenosine on the mitochondrial biogenesis in FRDA fibroblasts treated with BSO. Mitochondrial biogenesis was evaluated following pre-treatment with adenosine for 2 h and 12.50 mM BSO for 24 h. Data are expressed as mean ± SD and statistically analyzed by one-way ANOVA with Tukey’s post-hoc test. Hash (#) and asterisk (*) indicate significant differences (*p* < 0.05) in the SDH-A/COX1 ratio compared to the idebenone group and BSO-treated (0 µM adenosine) group, respectively. BSO, L-buthionine sulfoximine; COX1, cytochrome c oxidase subunit 1; SDH-A, succinate dehydrogenase subunit A.

**Figure 6 biology-12-00559-f006:**
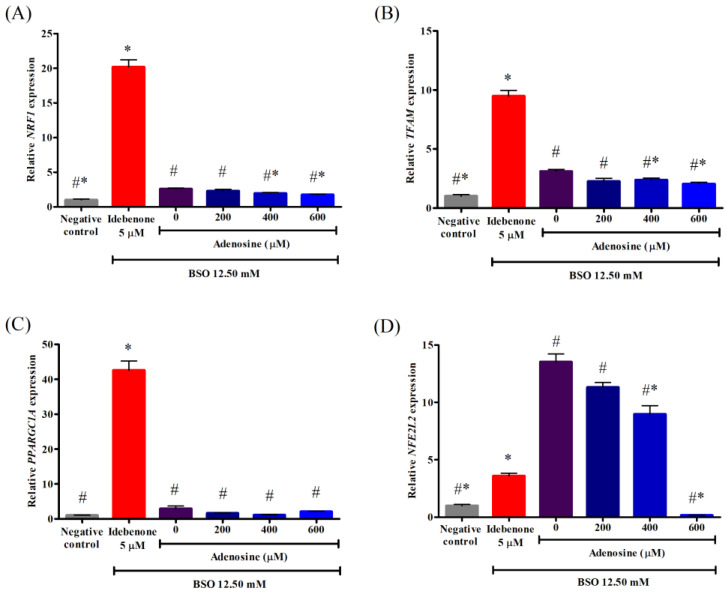
Effects of adenosine on the relative gene expressions associated with mitochondrial biogenesis in FRDA fibroblasts treated with BSO. Gene expressions of (**A**) *NRF1*, (**B**) *TFAM*, (**C**) *PPARGC1A*, and (**D**) *NFE2L2* were evaluated by RT-qPCR following pre-treatment with adenosine for 2 h and 12.50 mM BSO for 24 h. Data are expressed as mean ± SD and statistically analyzed by one-way ANOVA with Games–Howell post-hoc test. Hash (#) and asterisk (*) indicate significant differences (*p* < 0.05) in the relative gene expressions compared to the idebenone group and BSO-treated (0 µM adenosine) group, respectively. BSO, L-buthionine sulfoximine; *NFE2L2*, nuclear factor-erythroid 2-related factor 2; *NRF1*, nuclear respiratory factor 1; *PPARGC1A*, PPARG coactivator 1 alpha; *TFAM*, transcription factor A, mitochondrial.

**Figure 7 biology-12-00559-f007:**
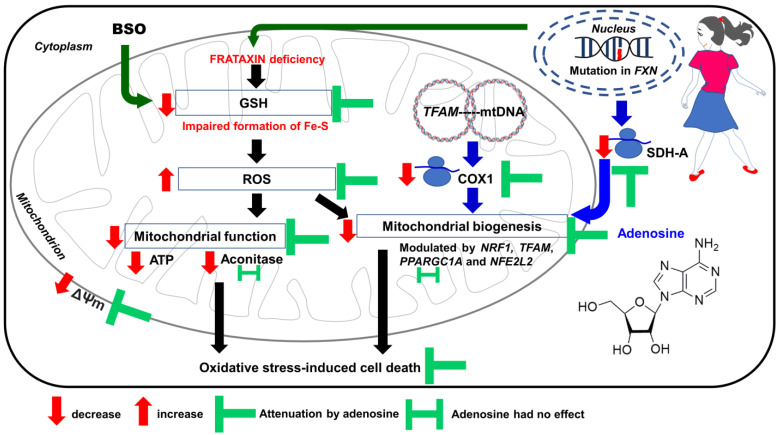
Proposed mechanism in which adenosine improves mitochondrial function and biogenesis in FRDA fibroblasts following BSO-induced oxidative stress. Illustration was created using Microsoft PowerPoint Professional Plus 2019. ATP, adenosine triphosphate; BSO, L-buthionine sulfoximine; COX1, cytochrome c oxidase subunit 1; Fe-S, iron–sulfur; *FXN*, frataxin; GSH, glutathione; mtDNA, mitochondrial DNA; *NFE2L2*, NFE2-like bZIP transcription factor 2; *NRF1*, nuclear respiratory factor 1; *PPARGC1A*, PPARG coactivator 1 alpha; ROS, reactive oxygen species; SDH-A, succinate dehydrogenase subunit A; *TFAM*, transcription factor A, mitochondrial; ΔΨm, mitochondrial membrane potential.

**Table 1 biology-12-00559-t001:** RT-qPCR primers.

Gene	Origin	5′–3′ Primer Sequence
*NFE2L2*	Human	Forward: ACA CGG TCC ACA GCT CAT C
		Reverse: TGT CAA TCA AAT CCA TGT CCT G
*NRF1*	Human	Forward: AGG AAC ACG GAG TGA CCC AA
		Reverse: TAT GCT CGG TGT AAG TAG CCA
*PPARGC1A*	Human	Forward: TTG ACT GGC GTC ATT CAG GA
		Reverse: GGG CAA TCC GTC TTC ATC CA
*TFAM*	Human	Forward: GTG ATT CAC CGC AGG AAA AGC
		Reverse: GTG CGA CGT AGA AGA TCC TTT C
*ACTB*	Human	Forward: GCC AAC ACA GTG CTG TCT GG
		Reverse: CTG CTT GCT GAT CCA CAT CTG C

*ACTB*, actin beta; *NFE2L2*, NFE2-like bZIP transcription factor 2; *NRF1*, nuclear respiratory factor 1; *PPARGC1A* PPARG coactivator 1 alpha; *TFAM*, transcription factor A, mitochondrial.

## Data Availability

Data sharing is not applicable. No new data were created or analyzed in this study. Data sharing is not applicable to this article.

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
