# Peer review of "Adenosine Improves Mitochondrial Function and Biogenesis in Friedreich's Ataxia Fibroblasts Following L-Buthionine Sulfoximine-Induced Oxidative Stress"

_biology, 2023, doi:10.3390/biology12040559_

Round 1

Reviewer 1 Report

The authors outline results of a study investigating adenosine in FDRA patient fibroblasts. The authors report beneficial effects of adenosine in the FDRA fibroblasts. The study is in an interesting area and has potential to further the case of testing adenosine in FA. However, several points need to be addressed by the authors. The first of which is that the entire manuscript is based upon 1 fibroblast line from 1 FA patient. Any responses seen could be specific to this fibroblast line, the work needs to be repeated across multiple fibroblast lines from multiple FA patients.

In addition, the Figures shown set the FA fibroblasts to 100% and look for the reduction in several parameters after treatment with BSO; there is no data in many Figures about any differences between control and FA fibroblasts, if the BSO has the same effect in the control and FA patient fibroblasts and if the adenosine has the same effect across the control and FA fibroblasts. 
The authors desrcibe the treatment effect for aconitase which are not there, no signficiant changes or even trends can be seen from the data presented.

The data on mitochondrial biogenesis is in the same format without any control fibroblast data what so ever. 

The authors have only done pre treatment experiments and have not done any treatment at the same time or post treatment assays to assess the real therapeutic potential of adenosine for FA.

In the discussion, the authors over state their results based upon data from 1 FA patient fibroblast line without other lines and other in depth mechanism assays.

Author Response

We would like to thank the editor and reviewer for careful and thorough reading of this manuscript as well as for the thoughtful comments and constructive suggestions, which help to improve the quality of this manuscript. Please find our response as attached:

Reviewer 2 Report

This manuscript reports a pharmacological in vitro characterization of adenosine protective action against oxidative stress in primary fibroblasts from Friedreich ataxic human donor, including biochemical antioxidant responses and expression of mitochondrial biogenesis markers. Since the main cause of deaths in Friedreich’s ataxia is cardiac failure relying on mitochondrial dysfunction, as well-documented in Introduction of this manuscript, this study brings the rationale for developing adenosine-based novel therapies which would improve patients’ fate in this genetic disease.

The manuscript is very well-written. The experimental approach in well-conducted and precisely described, regarding both experimental procedures and results. The detailed and wide range of assayed doses of adenosine brings originality with regard to previously published literature, and improves globally the characterization of adenosine action on oxidative stress in Friedreich's ataxia. The discussion is well structured and strongly documented, in particular regarding the doses of adenosine used in previous related publications. The concluding and summarizing scheme (Figure 7 and graphical abstract) is clear and useful.

However, a minor correction is required before the manuscript can be published (see below).

Scientific concern.

There is one missing issue in Discussion: to raise a hypothesis about the potential mechanism by which adenosine could trigger all the cytoprotective responses assayed in the present study. I.e. briefly, to discuss what might be the underlying nature of the blue inhibitory arrow from “Adenosine” to ROS genesis in Figure 7. Please authors add a final short paragraph in Discussion raising this issue, especially since the receptor-mediated mechanism of a presently used drug has been precisely recalled in Introduction (the A2A receptor antagonist istradefylline KW6002, lines 82-84).   

Author Response

(The authors gave the same response as above.)

Reviewer 3 Report

The manuscript titled "Adenosine Improves Mitochondrial Function and Biogenesis against L-buthionine Sulfoximine-induced Oxidative Stress in Friedreich’s Ataxia Fibroblasts" investigates the effect of adenosine treatment on mitochondrial function of the fibroblasts isolated from Friedreich's ataxia patients and treated with a very high dose of pro-oxidant. Friedreich's ataxia is a devastating genetic disorder, for which no cure is available. Therefore, research efforts trying to find new treatments for the disease are well justified.

Major concerns: However, I have concerns about the model the authors use. Friedreich's ataxia is associated with mitochondrial impairment, and therefore the fibroblasts from Friedreich's ataxia patient would be expected to be more sensitive to pro-oxidant treatment. However, the authors are using a very high dose of glutathione synthesis inhibitor to see a decrease in mitochondrial membrane potential and ATP production. It would be important to know why the authors chose a so high concentration of pro-oxidant and how the control fibroblasts react to it. Do the authors see any differences in mitochondrial function between control and diseased fibroblasts in the absence of pro-oxidant treatment? Also, adenosine doses the authors use are very high. It is not clear why the authors did not try lower doses. It would also be important to determine through which receptors adenosine would exert its effect on fibroblasts. The use of idebenone as a positive control is also confusing. It has been published previously that idebenone induces mitochondrial depolarization and a dramatic decrease in ATP production, eg 10.1016/j.bbabio.2011.10.012.

Minor concerns:

1. The graphs are shown at poor resolution and there are some asymmetries in the graphs so that it looks like they were drawn by hand. In Figure 2, 600 uM adenosine bar does not have an error bar at all. 

2. It is not appropriate to write in Figure legends "analyzed by Tukey's test". It would be more appropriate to write: "analyzed by one-way ANOVA with Tukey's post hoc test". 

3. Line 56: chromosome location should be changed to 9q21.11.

4. Line 80-81, it would be logical to change the order of Huntington's and Alzheimer's disease here.

5. Line 86: "There is scant evidence showing that adenosine has protective effects on cerebellar ataxia and the mechanisms are not clear". It would be good to add references here.

6. Line 152, the reference should be formatted correctly.

7. Figure 7 is very confusing. It suggests that adenosine treatment promoted NRF1, Nrf2, and TFAM expression. In reality, adenosine treatment decreased the expression of these genes in comparison to  BSO alone, at least at the time-point measured. The authors could consider an earlier time-point for RT-qPCR samples.

8. In the graphs, I would change the order of Idebenone and BSO-only graphs putting BSO-only next to adenosine-treated bars and marking it as 0 uM adenosine. 

Author Response

(The authors gave the same response as above.)

Round 2

Reviewer 3 Report

Dear authors, thank you for your corrections and clarifications! I would recommend publishing the manuscript after the gene names are corrected to the standard ones. For example, the gene name for Nrf2 is NFE2L2

Author Response

Dear Prof/Dr,

Thank you for your constructive feedback and suggestion. We have amended the gene names according to the HUGO Gene Nomenclature Committee. 

ACTB  actin beta
https://www.genenames.org/data/gene-symbol-report/#!/hgnc_id/HGNC:132
  FXN - frataxin https://www.genenames.org/data/gene-symbol-report/#!/hgnc_id/HGNC:3951   NFE2L2 - NFE2 like bZIP transcription factor 2
https://www.genenames.org/data/gene-symbol-report/#!/hgnc_id/HGNC:7782
  NRF1 - nuclear respiratory factor 1
https://www.genenames.org/data/gene-symbol-report/#!/hgnc_id/HGNC:7996   PPARGC1A PPARG coactivator 1 alpha 
https://www.genenames.org/data/gene-symbol-report/#!/hgnc_id/HGNC:9237   TFAM - transcription factor A, mitochondrial 
https://www.genenames.org/data/gene-symbol-report/#!/hgnc_id/HGNC:11741   Wong Kah Hui